# The Potential Use of Sex Robots in Adults with Autistic Spectrum Disorders: A Theoretical Framework

**DOI:** 10.3390/brainsci13060954

**Published:** 2023-06-15

**Authors:** Fabrizia Pasciuto, Antonia Cava, Alessandra Falzone

**Affiliations:** Department of Cognitive Sciences, Psychology, Education and Cultural Studies (COSPECS), University of Messina, 98121 Messina, Italy; antonia.cava@unime.it (A.C.); alessandra.falzone@unime.it (A.F.)

**Keywords:** autism, sex robot, sexuality

## Abstract

Although the importance of the sexual sphere for the health of all human beings has been recognized at an international level, often this is underestimated when it comes to disabilities and even more to intellectual disabilities. In fact, the idea that subjects with intellectual disabilities are not aware of their bodies and of their wishes in the sexual and emotional field is still widespread in our society, in such a way that they are considered as children in need of constant supervision. Moreover, further hints of criticism that can be raised are about the poor level of sexual education that is dedicated to these subjects, both by family members and by therapists. The last decades have been characterized by a considerable growth in the technological sector and many new instruments have been successfully used in the field of healthcare of weak or disabled subjects. A particularly fruitful branch has been robotics which, in subjects with autistic spectrum disorders (ASD), has revealed itself as an excellent support to stimulate communication and develop social skills. As in recent years the field of robotics has also been characterized by a strong interest in the sphere of sexuality, building and implementing what we now define as sex robots or sexbots, it could be interesting to start a debate on the potential that these new generation artificial agents could have in the field of care of subjects with ASD. These robots, possessing a technology based on stimulating verbal and nonverbal interaction, could be useful for an education that is not only sexual but also psycho-emotional in subjects with ASD.

## 1. Introduction

The current sociocultural context characterizing the world we live in is in many aspects dominated by stereotypes and prejudices which, within the framework of disabilities, tend to create an almost dichotomous distinction between disability and sexuality/affectivity, placing them in a position that seems almost incompatible. 

In 2007, the UN convention [1] on the rights of people with disabilities and the relative optional protocol was drawn up. The convention lies within the framework of protection and promotion of people’s rights and dignity and, with specific reference to the sexual sphere of the individual with disabilities, foresees that these subjects have the right to take advantage of the same services provided to other individuals—including healthcare services in the sexual area—and acknowledges respect for private life and the right to set up a family. However, it must be stated that, in spite of harsh debate and of the efforts by several public and private bodies, there is still some resistance in recognizing and talking about sexuality in disabled subjects. It is not rare, in fact, to identify the barriers still existing in front of these subjects, whose sexual and emotional development has often been impeded by obstacles and hindrances: physical barriers making some spaces inaccessible, lack of strictly sexual and emotional education, or of socialization with subjects outside their family or healthcare environment. 

In particular, so-called sexual and parenting rights (SPRs) concern fundamental rights such as the right to marriage, family, parenting, relationships, fertility, access to information, and sexual and reproductive health services. 

As we see in detail in this paper, disabled people are often seen in a stereotypical way and their sexual needs are ignored or hidden. 

For instance, a study conducted on a sample of the Italian population investigated the level of agreement with the rights of people with physical disabilities and the rights of people with intellectual disabilities. While a higher proportion of Italians agreed with the SPRs of individuals with physical disabilities, it was quite different for those with intellectual disabilities. In fact, almost 80% were against rights such as adoption and were strongly influenced by stereotypes [2].

The panorama of disability, moreover, is characterized by a very high level of heterogeneity, which should push us to use the term disability in its plural declination and to recognize the different social and cultural contexts in which it is framed. 

## 2. Disability Studies

Disability Studies are a relatively recent discipline born with the aim of analyzing disability in the social, political, historical, and cultural panorama. Historically, this branch of research started in the UK around the 1970’s. More specifically, Shakespeare identified 1975 as “Year Zero” marking its birth with the draft of the Fundamental Principles of Disability by UPIAS, The Union of the Physically Impaired Against Segregation [3]. Disability Studies focus on the need to consider disability not just from a biological point of view, but in its complexity and above all within the social sphere. This is considered as the first cause of oppression and carrier of discrimination and exclusion of subjects with disabilities, often confined in special places of education, work or socialization [4,5]. The new feature deriving from the axioms of this social theory of disability is essentially to entrust to medicine the problems related to body but, at the same time, considering them as a social issue, whose objective is a radical change of society in its relation to disability [6,7]. It is needed to overcome the stereotypes in which, day by day, human beings are placed to finally state that discrimination, in all its forms, is unacceptable and must be overcome. 

The starting point for the social theory of disability is an analysis that is not only sociological, but also historical and anthropological, necessary to understand how disability is framed within a wider cultural panorama, which has led scholars to state that it is socially structured and culturally produced [6]. It is exactly the culture of a society that influences the way in which disability is perceived, whether it is physical or mental. 

Therefore, disability studies distinguish themselves from the medical paradigm in terms of some specific characteristics. While medicine aims at disabled subjects’ reintegration through their rehabilitation, the social theory of disability is not based on fighting and eliminating individuals’ shortcomings, but on their reintegration by fighting against such discriminations as high unemployment rates, insufficient social and economic policies, and architectural and cultural barriers. All these aspects, if not adequately contrasted, cause disabled subjects’ isolation, making them dependent on third parties and excluded from society. In this way, the concept of disability is in a certain sense redesigned, putting forward the hypothesis according to which it is exactly society that makes a subject disabled, by enacting that process defined as disablement and categorizing those who are different from what is considered as the standard. Therefore, there is a passage from the individual to the social and consequently the disadvantage that is usually attributed to the disabled subject does not represent any more a deficiency deriving from a difference in biological sense, but the result of particular social processes. The traditional dichotomy standard/deficit collapses, instead shedding more light on the debilitating conditions caused by society itself. 

Disability studies also analyze disabled subjects’ sexual sphere, which is often ignored and denied as their body is considered asexual and unattractive [8]. From this perspective, many associations have claimed disabled subjects’ rights, especially those with intellectual disabilities, to have their own sexual independence and the right to decide for their own fertility [9,10]. It is also evident that those approaches investigating sociopolitical structures and cultural barriers still limiting disabled subjects’ free sexual expression and the opportunities they have are of crucial importance [11,12]. Research suggests, in fact, that when disabled subjects’ sexual sphere is investigated, it is possible to find several obstacles posed by public bodies or entities, limiting action. A tangible example is represented by the figure of sexual assistants who, in spite of possibly representing a strong help for people with some disabilities, are obliged to depend on the policy of the state they are in, which often limits possibilities, assimilating these jobs to forms of prostitution.

## 3. Sex as a Taboo

Although we live in an age when sexual references are ubiquitous, it is also paradoxically impossible to deny that sex is still seen as a taboo in our society and, above all, it is so when referring to disabled subjects [13]. According to Shakespeare, in fact, this perception about sexuality puts disabled people in a position of anguish and insecurity, therefore it is easier to hide this sphere of life and act as if it does not exist [14]. Against a constant sexualization of bodies on the media, in advertisement and in most of what surrounds us, disabled bodies are continuously hidden as if their peculiarities do not deserve attention. 

The term erotophobia has been coined to define fear linked to the sexual sphere. In the specific field of disability studies and of those movements fighting for minorities’ rights, this term has been used to define the so-called negative sexual approaches causing discrimination and oppression: “Erotophobia involves not only explicit declarations of pathology, but also other practices and attitudes that more subtly reflect cultural taboos against sexual practices, desires, and identities” [15]. 

Therefore, it seems that the message often conveyed to disabled subjects is aimed at stating that their sexuality is something unappropriated. Especially in the field of intellectual disabilities, there is the belief that these subjects must be considered as children needing support and therefore lacking the capacity to make decisions in the sphere of sexuality [15]. 

This taboo is connected to the idea that they cannot have conventional relations and therefore their only possibility to have a sexual relation is by paying. However, it is also right to state that often the isolation they suffer from makes it possible to see in sex work one of the few opportunities to meet their needs. The survey “Time to Talk Sex” by Disability Now carried out in 2005 showed that 38% of disabled men and 16% of women have considered the opportunity to pay for sex and, while 22% of men have declared to have actually taken advantage of these services, only 1% of women have done so [16]. Moreover, it has been highlighted that due to cultural barriers in our society disabled subjects do not visit a sex worker exclusively to meet their sexual needs, but also just to create intimacy and interpersonal relations [14]. 

A more distinct interest for sexual sphere and human rights has its roots in the movements born in the last century [17], whose members started to think about a sexual “subjectivity” including several concepts such as identity and sexual orientation, sexual desires, gender identity, and sexual practices. Therefore, acknowledging and working on sexual rights poses a series of challenges for our society, designing new strategies to claim them [18]. 

It is in fact possible to state that, apart from particular congenital anomalies of genital organs, the characters representing sexuality evolve in the disabled subject in a way that is almost identical to most of the population. For this reason, it is necessary to guarantee these subjects the possibility of living their sexuality as naturally as possible. 

## 4. Sexuality in Intellectual Disability

Generally, sexuality is defined as a set of behaviors including social, emotional, and physical interactions, not restricted to sexual relations [19]. However, when we refer to subjects with intellectual disabilities (ID), prejudices and stereotypes related to the sphere of sexuality become particularly strong. In fact, the belief that these subjects do not have the same urges as most of the population and that their personality and awareness of their body can be compared to a child’s is widespread. This simplistic classification can cause social and psychic discomforts in the subject, who is often in a condition confined outside not only sexual education, but also emotional and relational [20]. Therefore, there is often a paternalistic approach to subjects with intellectual disability, restricting the expression of their sexuality and, moreover, not recognizing it as expressed in different forms, going beyond mere sexual relation. Therefore, it is exactly this hyperprotection that limits these subjects’ growth of awareness in the sexual field [21,22]. 

A rethinking of policies referring to disabilities and the sexual sphere could be useful, in sharp contrast with those conceptions still identifying the subject with intellectual disability as an asexual/hypersexual personality [23], not interested in emotional relations, a victim, or unable to recognize his/her own desires. 

Cared for by parents or caregivers even in adult age, people with intellectual disability are often unable to start social relations, friendships, or emotional relations [24]. Some research carried out in the field of sexuality and referring to the relation between parents and teenagers with intellectual disability has shown how most of these families have several difficulties in facing topics related to sexual and emotional education [25,26]. 

With specific reference to subjects with autistic spectrum disorders, it is useful to bear in mind that subjects with ASD have individual urges needing a sexual education different from other subjects with intellectual disability [27]. Moreover, several pieces of research have demonstrated that these subjects’ sexual desires are substantially equivalent to those of non-autistic adults [28]. For example, a study carried out on teenagers in an age range between 15 and 18 years old highlighted that the behavior of teenagers with high functioning autism are more or less the same as the general population as far as the sphere of sexuality is concerned. The result is therefore that 94% of teenagers with ASD have had sexual behaviors on their own and this datum, if compared to non-autistic subjects, is a percentage pointing out high similarities [29].

Notwithstanding this, these subjects’ interpersonal relations are limited by communicative difficulties characterizing ASD subjects and low chances of meeting other people [30,31], but also by poor sexual education proposed both by society and—often—by the family [32,33]. Not receiving adequate sexual education above all during teenage years, in adult age, these subjects find themselves lacking fundamental knowledge to understand and communicate their needs and desires, the functioning of their body, but are also unaware of their rights and of the possibility to access resources made available for sexual and reproductive health [34]. 

As far as these difficulties are concerned, it is rightfully urgent to rethink about the figure of the sexual assistant. Sexual assistance has been defined as: “a sexual accompaniment service for people with disabilities that provides educational services about sexual practices and support services for sexual activity with the aim of meeting clients’ sensual or sexual needs while bearing in mind the specific characteristics related to their disabilities” [35]. The figure of the sexual assistant, however, is often victim of a controversial situation as in many countries it is assimilated to prostitution, and it is consequently forbidden to exercise this profession [36]. It is crucial to state that there are substantial differences between prostitution and sexual assistance and, in many of the countries in which this profession is not regulated, several movements have been created in support of its approval [37]. Professionally, moreover, the sexual assistant has the obligation to attend a training course to gain in-depth knowledge of disabilities and deal with the difficulties that each of them can cause [38]; moreover, in most cases, the sexual assistant works under the supervision of a third party represented by a therapist [39]. 

Taking a perspective that looks at the sexual assistant as an extremely positive profession for overcoming taboos related to disabled subjects’ sexuality and for a good sexual education, especially in subjects with ASD, it is possible to conclude that experimenting eroticism and affectivity is one of the key points to understanding their body and having a free and conscious sexuality. These experimentations could also take advantage of particular and new robotic technologies that, in the very near future, could involve more and more people: the use of sex robots [39]. 

## 5. Sex Robot Characteristics 

A sex robot can be defined as an artificial agent designed to be used for sexual or recreational aims. To understand its physiognomy, it is desirable to set the basis with the definition proposed by John Danaher, who pointed out the three necessary characteristics to be able to define them: a sex robot must possess a body of humanoid form; have typically human behavior and movements; and possess a system of artificial intelligence of relatively advanced level [40]. It is exactly the possession of these three factors that makes it possible to distinguish a sex robot from the most common sex toys and especially from sex dolls. 

Therefore, taking Danaher’s perspective, it is possible to identify only few prototypes meeting these needs in the current market. 

Chronologically, the first sex robot was introduced to the market in 2010 when, during the AVN Adult Entertainment Expo in Las Vegas, the company True Companion, represented by Douglas Hines, showed the audience its prototype: Roxxxy. The robot had a humanoid body with synthetic leather, knew its name and could speak and listen to its interlocutors. Aesthetically, it could be personalized at convenience. The launch of Roxxxy was immediately considered a success and in very short time the company received about 4.000 preorders. However, still today, it seems that the sex robot by True Companion has never really been traded and many are skeptical about its real production. 

Already existing on the market and a leader in the sector is the company Abyss Creation, born from an idea by the sculptor Mattew McMullen. The first sex robot by Abyss Creation was launched on the market in 2017 with the name Real Doll X and the debut of Harmony. This new sex robot has established itself as unique in its genre. Thanks to the software XMode Harmony it can rotate its head, reproduce facial expressions through the movement of eyes, eyebrows, and lips and it can have real conversations. Aesthetically, Real Doll X can also be totally personalized both in details of face and body. Real Doll X possesses such a technology that allows its body to have a temperature similar to men and, thanks to several sensors spread along its body, it can recognize when it is touched and react consequently. Among the most revolutionary functions, there is also the possibility to connect the sexbot to an app allowing users to assign it the kind of personality they want, choose among four different types of voice and create an avatar. 

## 6. Pros and Cons of the Spread of Sex Robots

The more and more massive development of categories of robots interacting with human beings in ways that tend to increasingly access our intimacy has created a situation in which, in the literature, there are now dichotomic positions both on their usefulness and on their moral lawfulness (Figure 1). Among those who believe that the spread of sex robots could be positive for our society, the figure of Levy stands out. An expert in artificial intelligence, Levy aims at explaining how, in some decades, human beings will perceive robots as particularly attractive partners, thanks to these artificial agents’ ability to increasingly show talents and skills [41]. From Levy’s reflections, it is pointed out how, over the next decades, having a sex robot would not be considered any more extravagant or even morally wrong as it is today. Over the years, in fact, society has radically modified its conceptions about sex and it is possible that this is also going to occur for these new artificial partners. Levy seems very optimistic about the spread of this new family of robots, even stating that the benefits for our society could be several: from the reduction in pregnancies during teenage years to a lower spread of sexual diseases, even to management of pedophilia. Never forgetting the moral considerations at the basis of the debate, Levy draws his conclusion: probably, by 2050, we will live in a world where there will be no more a strict distinction between biological and artificial creatures, it will be like entering into contact with a new population up to that moment unknown and, above all, able to meet every desire, not only in the sexual, but also in the emotional field.

Following this same line of thinking are the considerations put forward by McArthur [40], who first of all analyzed the issue of privacy, intended as the possibility to have relations with a robot in one’s own intimacy and, above all, without implying any kind of direct damage to other people. It is evident that accessing these people’s private life would represent a limitation on their freedom. However, it is also true that individual rights are never absolute and, above all, the impacts that sexbots can have on society must be considered. Alongside this, McArthur also underlines the importance of moral judgements as, if many think that an individual’s privacy cannot be violated in the abovementioned cases, often this imperative collapses when morality is at stake. It is in fact not rare that judgements are expressed about the moral lawfulness of some actions when these take place in one’s own private sphere. In conclusion, according to the scholar, the spread of sex robots must be guaranteed by the right to privacy, and therefore it is not possible to declare that it must be tolerated, but it should rather be actively encouraged to overcome arising social stigma. 

Considerations about the introduction of sex robots in our houses, however, are not always in favor. In a diametrically opposed perspective in comparison to what has been pointed out so far, there are authors such as Richardson, founder of the Campaign Against Sex Robots (CASR), strongly supporting the idea of sex robots as desubjectivating technological artifacts and carriers of inequalities [42]. For Richardson, supporter of a perspective putting sex robots and exploitation of prostitution on the same level, the spread of these new technologies would represent a considerable loss of sympathy for women, in this way making them objectified and dehumanized. Therefore, diametrically opposing McArthur, Richardson thinks about it as a public issue, pointing out the risks of violence that women and children can run if objects of these desires of submission. Equally, Sullins states that the implementation of sex robots contributes to creating an image of the female body that is stereotyped and exaggerated, totally faking it [43]. 

Another difficult issue is consensus, on which Gutiu has widely expressed her views [44]. The scholar underlined how, in contemporary society, there are still considerable power imbalances between men and women and these imbalances are replicated every time a woman-like sexbot is created. It is exactly through the perpetuation of gender stereotypes that these technologies are able to avoid consensus of the subjects involved in these relations, and in this way, they emulate female sexual slavery. It is also important to point out that there are no male-like sex robots available on the market at the moment. For this reason, the literature against the introduction of these robots into our society has focused on the issue of a highly stereotyped and sexualized image of women. 

Many studies have shown that the sex robot market is targeted almost exclusively at male consumers [45,46,47]. Furthermore, our society often has a limited view about female sexuality and reduces it to the procreational sphere. A survey on sexually explicit media (SEM) [48] showed that the way sex robots are designed and promoted promotes unreal beauty standards and sexual performance, and this may provoke anxieties and fears in women. It could be beneficial to produce and propose sex robots designed with a stronger female-centered vision. Leaving the stereotypes that guide our society and applying a truly inclusive approach in the design of sex robots could shift the market through the inclusion of many different individuals.

Not intending, in this paper, to assume in full either of the positions just mentioned, we believe it is necessary to conduct research on sex robots by taking a critical view, without demonizing or fully extolling the production of such robots, identifying their criticalities and strengths. These reflections are necessary in order to identify and attempt to prevent degenerations such as increases in violence or isolation and alienation of users. As long as there is always consent on the part of the partner, there is no reason to consider some sexual tastes or practices more moral or licit or better than others. So, we should question why having sexual intercourse with a robot should not be allowed.

**Figure 1 brainsci-13-00954-f001:**
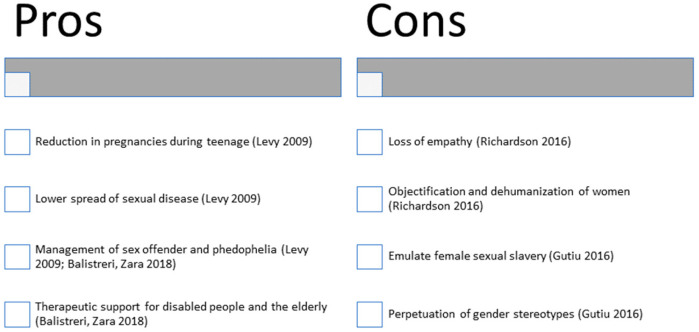
Pros and Cons of Sex Robots [40,43,44,49].

## 7. Sex Robot Application for Healthcare in Adults with Autistic Spectrum Disorder

More hints and proposals for implementation also come from Balistreri, who proposes an in-depth analysis on the different views about sex robots and is in favor of a massive introduction of these artifacts in our houses not only as sexual partners but also as therapeutic supports [49]. To demolish the widespread reluctance in many people when it comes to sex robots, it is first of all necessary to eradicate prejudices about moral lawfulness of some sexual tastes. Naturally, for obvious reasons, it is unacceptable to have sex with somebody against their will as well as any other form of violence but, from a very different perspective, sex between consenting people must be accepted also if their sexual tastes cannot be understood by everyone or even considered eccentric or repellent. Therefore, for some people, having a sexual relation with a robot could look like an extravagance but it should not be condemnable for this reason.

Although not empirically used or studied yet, sex robots have been proposed as a support in the field of healthcare of both weaker subjects, for example, the elderly, often excluded and marginalized as far as the sexual sphere is concerned [50], and of those who have to deal with arising diseases or disabilities.

In this perspective, it could be interesting to use sex robots for an education that is not only sexual but also psycho-emotional in subjects with ASD.

Involving robotics in the healthcare sector is nothing new; in fact, what has been defined as socially assistive robotics (SAR) focuses on robotic agents equipped with the necessary skills to also interact in a social way (as the adverb socially suggests) with human beings and therefore be able to provide care to people with special needs connected to their age [51], arising diseases, or disabilities [52]. It is, therefore, an investigation field focused on the target of providing healthcare in physical and mental health through social interaction, associating support in rehabilitation, or in learning from a psychological support. As far as disorders in the autistic spectrum are concerned, they affect abilities such as communication, learning, imagination, and interaction with other people, sometimes leading subjects to isolation from their social environment. For this reason, subjects with autism often show difficulties in relations, in verbal and nonverbal communication, and the tendency to prefer, with their interlocutors, simple, repetitive, and highly predictable patterns of behavior. Many researchers, in fact, have demonstrated that robots can represent a very useful therapeutic instrument for autism, benefiting from recurrent interactions with social tracts between subjects and robots [53]. To better try and meet these subjects’ therapeutic needs, scholars have believed it necessary to create a highly safe and predictable environment of interaction, so that robots can promote spontaneous activities of interaction. At the heart of these scenarios, there are focusses such as stimulating interaction, maintaining attention, and stereotyped behaviors [54]. 

Within this field of application, it could therefore be positive to use sex robots as therapeutic supports, under the supervision of third parties represented by psychologists or therapists, to stimulate sexual and emotional education in subjects with autism. In fact, as far as the sphere of sexuality is concerned, robotic platforms have not been used yet. Among the models previously presented, the robot created by the company Abyss Creations and belonging to the category of RealDoll X stands out for functionality and quality of performances. Equipped with artificial intelligence making it able to have more or less complex conversations, adapting to users and learning from their own experience, using sex robots could be beneficial not only for a mere experimentation of erotic acts, but also to stimulate the emotional–sentimental dimension that is often denied to subjects with ASD. Thanks to their functions, it could be possible to use these sex robots along a path that has its basis, first of all, in real sexual education, and later in a process of discovery of one’s own personal sexuality by developing deeper knowledge of one’s body and of the sensations that one can feel. Furthermore, as already explained in the previous paragraphs, subjects with ASD often tend to live in a social environment where interpersonal relations are very limited, and this can cause isolation. Additionally, one more factor to take into consideration regards parents or caregivers’ difficulties, as they often find themselves unprepared to welcome the changes occurring firstly during puberty and the impulses arising during the adult age of an individual with ASD. These needs can be not exclusively sexual but are often accompanied by a desire for emotional and sentimental relations. Therefore, meeting these needs does not exclusively end with the mechanical act of sex but becomes part of a prism full of facets. Social interaction with a sex robot could therefore represent a stimulus to a sexual education both theoretically going through the topics of sexuality, affectivity, and eroticism but also aiming at stimulating communication between a subject with autism and a possible relationship with another individual. Taking this perspective, robots could be considered first of all as tutors, teaching them to know about themselves and about others. 

## 8. Conclusions

The possible use of sex robots in the field of healthcare is still at an early stage. However, it is undeniable that in order to implement the quality of life of subjects with autistic spectrum disorder, it is necessary to guarantee all the rights they need, including those within the sphere of sexuality. In this paper, we discussed the contribution that sex robots could have in the lives of subjects with ASD, potentially bringing benefits through sexual and psycho-emotional education carried out under the supervision of a specialist. 

It is of course undeniable that, as with all new technologies launched on the market, even more for those intruding on our intimacy, in-depth ethical and moral considerations cannot be ignored, always bearing in mind the profound impacts that these can have on our lives.

## Data Availability

Data sharing not applicable.

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
