# Peer review of "The Potential Use of Sex Robots in Adults with Autistic Spectrum Disorders: A Theoretical Framework"

_brainsci, 2023, doi:10.3390/brainsci13060954_

Round 1

Reviewer 1 Report

This article discusses the potential use of sex robots and their technology as a therapeutic option/support in patients with ASD. This is a refreshing and timely article that I genuinely enjoyed reading. It is exceptionally well written and touches upon many different aspects of the topic, clearly demonstrating that the authors performed a thorough literature research and have a deep understanding of the topic. I have nothing to criticise except the fact that a Figure illustrating the key points and hypotheses, as well as pro and cons of using sex robots, would be extremely helpful and appreciated. I believe that this would drastically improve the quality of the article, making it more attractive to a broader readership.

Author Response

Dear Reviewer, 

I am writing to express my sincere gratitude for your valuable feedback and guidance throughout the review process of my paper. Your constructive comments have contributed to the improvement of my work. I am pleased to inform you that I have considered your suggestions regarding the inclusion of Figures in my paper. Recognizing the significance of visual aids in enhancing the clarity and understanding of the research findings, I have incorporated the recommended images into the revised version of the paper. I believe these additions will significantly strengthen the visual representation of my research and provide better support for the readers.

Thank you once again for your review.

Reviewer 2 Report

I would like to thank the authors and the Editorial Board for the opportunity to review the article submitted to MDPI’s Brain Sciences. The authors' manuscript refers to a very important topic: the use of robots in sexual assistance in individuals with autistic spectrum diagnoses.

 I find the authors’ manuscript very interesting and in line with my clinical practice. I’m a psychologist and sexologist who works with ID and ASD individuals. I agree with the authors that the sexuality of those individuals is still a taboo subject, not receiving enough attention from clinicians and researchers. In regards to the authors’ manuscript, I have only three comments. I believe that two minor changes have to be done so the manuscript meets all MDPI guidelines. I have only one major concern and I would like the authors to elaborate on that issue.

Two minor issues

Affiliation: please specify the authors’ affiliation. The University is not enough to verify the author’s expertise and field of research. As per MDPI’s guidelines: The PubMed/MEDLINE standard format is used for affiliations.

Contributions: contributions presented in the authors’ manuscript are not in line with CRediT taxonomy adapted by MDPI’s journals.

One major issue

The dangers and/or cons of sexbots usage: The authors refer to various references, but I find that the manuscript does not present their own judgment. It lacks critical analysis. The authors refer to various cons of sexbot usage, such as objectification of individuals or imbalances between men and women. For example, the authors refer to the cons of women-like sexbots, but the same cons can be applied to men-like sexbots. This section definitely needs additional thorough analysis.

Due to a lack of information on the authors’ affiliation (department, institute etc.) or their clinical work, it’s hard to evaluate if they have experience in working with ASD or ID individuals outside of the research field. If not, I still invite them to elaborate and discuss it with me on that matter nevertheless. One of the major aims of the rehabilitation and support of those individuals is to help them feel accepted and connected to their environment/society. Sexual rehabilitation with certified and trained sexual assistants would be more beneficial because it would be more humanizing. Sexual rehabilitation is needed, but the usage of artificial sources of sexual stimuli can be dangerous in the rehabilitation process because it would exclude them even more from society. I invite the authors to elaborate on that issue in bigger detail.

Author Response

Dear Reviewer, 

I am writing to express my appreciation for your valuable feedback and guidance during the review process of my paper titled. Your expertise and constructive comments have played a crucial role in refining my work.

I wanted to personally inform you that I have carefully considered and implemented the suggestions you provided.

First of all, I wanted to inform you that I have addressed the issue regarding the affiliation of the authors in my paper, as per your suggestion. I apologize for any confusion or oversight caused by the previous version.

Moreover, authors jointly conceived of this paper, drive by a collaborative discussion about the “the use of robots in sexual assistance”. All the authors wrote parts of this paper. Specifically, F.P wrote Sexuality in Intellectual Disability, Sex Robot Characteristic, Pros and Cons about the spread of Sex Robots, Sex Robots application for healthcare in adults with autistic spectrum disorder; A.C. wrote Disability Studies and Sex as a taboo; and A.F. wrote the introduction and the conclusion. All authors have read and agreed to the published version of the manuscript.

The emphasis on women in this research is not to exclude men or prioritize one gender over another. The research does not mention male-like sex robots because they have not yet been developed and commercialized. I have now included this detail in the text and the issue will seem clearer.

Regarding your last advice, we consider that interacting with the environment and society plays a crucial role in the development and well-being of those individuals. Emerging technologies, such as robots, hold the potential to provide valuable assistance and support in various aspects of their lives. In particular, robots can play a significant role in facilitating educational experiences for individuals with disabilities, offering opportunities for enhanced learning and empowerment. Especially Sex Robots can be programmed to adapt and customize affective and sexual educational content and methods. By working alongside individuals with disabilities, robots can encourage self-confidence and independence. During interaction between Sex Robots and those individuals, psychologists and sexologists can help individuals develop realistic expectations, cope with potential challenges, and maximize the benefits of these interactions. By providing psychological support, they can ensure that individuals feel comfortable, empowered, and informed during their interactions with those robots. You can find all these observations highlighted in the last paragraph and in the conclusion of the paper.

Furthermore, as mentioned, in countries such as Italy, despite being very significant, the work of the sexual assistant is not recognised by legislation.

Thank you once again for your review.

Reviewer 3 Report

Dear author,

Your article about “The potential use of Sex Robots in adults with Autistic Spectrum Disorders: a theoretical framework” fascinated me very much. Minor concerns are connected to the actual article, so I propose to submit it again with minor revisions.

The title is good. The abstract is detailed and in general, it’s good.

The reported results are interesting and clearly summarized.

Moreover, I suggest you cite an articles that can add interesting data to your introduction on perceived sexual rights for disabled people: Di Santo SG, Colombo M, Silvaggi M, Gammino GR, Fava V, Malandrino C, Nanini C, Rossetto C, Simone S, Eleuteri S. The Sexual and Parenting Rights of People with Physical and Psychical Disabilities: Attitudes of Italians and Socio-Demographic Factors Involved in Recognition and Denial. Int J Environ Res Public Health. 2022 Jan 17;19(2):1017. doi: 10.3390/ijerph19021017. PMID: 35055837; PMCID: PMC8775460.

Author Response

Dear Reviewer, 

I am writing to express my gratitude for the reading advice you provided for my paper's. By incorporating your recommended article, I believe the introduction now sets a stronger foundation for the subsequent sections of the paper.

I look forward to incorporating any further feedback you may have during the revision process. If there are any specific areas or aspects of the paper you believe require additional attention, please do not hesitate to let me know.

Thank you once again for your review.
